# Flow Cytometry: The Next Revolution

**DOI:** 10.3390/cells12141875

**Published:** 2023-07-17

**Authors:** J. Paul Robinson, Raluca Ostafe, Sharath Narayana Iyengar, Bartek Rajwa, Rainer Fischer

**Affiliations:** 1Department of Basic Medical Sciences, Purdue University, West Lafayette, IN 47907, USA; naraya94@purdue.edu; 2Weldon School of Biomedical Engineering, Purdue University, West Lafayette, IN 47907, USA; 3Molecular Evolution, Protein Engineering and Production Facility (PI4D), Purdue University, West Lafayette, IN 47907, USA; rostafe@purdue.edu; 4Bindley Bioscience Center, Purdue University, West Lafayette, IN 47907, USA; brajwa@purdue.edu; 5Department of Comparative Pathobiology, College of Veterinary Medicine, Purdue University, West Lafayette, IN 47907, USA; rainer-fischer@purdue.edu; 6Purdue Institute of Inflammation, Immunology and Infectious Diseases, Purdue University, West Lafayette, IN 47907, USA

**Keywords:** fluorescence-based cytomics, spectral flow cytometry, multiomics, single-cell analysis, microfluidics, high-throughput analysis

## Abstract

Unmasking the subtleties of the immune system requires both a comprehensive knowledge base and the ability to interrogate that system with intimate sensitivity. That task, to a considerable extent, has been handled by an iterative expansion in flow cytometry methods, both in technological capability and also in accompanying advances in informatics. As the field of fluorescence-based cytomics matured, it reached a technological barrier at around 30 parameter analyses, which stalled the field until spectral flow cytometry created a fundamental transformation that will likely lead to the potential of 100 simultaneous parameter analyses within a few years. The simultaneous advance in informatics has now become a watershed moment for the field as it competes with mature systematic approaches such as genomics and proteomics, allowing cytomics to take a seat at the multi-omics table. In addition, recent technological advances try to combine the speed of flow systems with other detection methods, in addition to fluorescence alone, which will make flow-based instruments even more indispensable in any biological laboratory. This paper outlines current approaches in cell analysis and detection methods, discusses traditional and microfluidic sorting approaches as well as next-generation instruments, and provides an early look at future opportunities that are likely to arise.

## 1. Background

For over 100 years, scientists have desired to analyze single cells based on a variety of approaches. As early as 1904, Kohler utilized fluorescence microscopy for the analysis of epidermal cells of salamander maculosa larvae [1]. This is considered to be the first use of fluorescence microscopy, the pioneering use of a fluorescence microscopy marker, and a departure from traditional spectrophotometric analysis. These experiments represent the beginning of the field of cytometry. Twenty years later, in 1924, the first detection and analysis of DNA was in experiments carried out by Robert Feulgen in what became known as the Feulgen reaction [2]. The principle of this reaction was the formation of Schiff’s reagent from pararosanilin and its reaction with aldehydes to form colored products. In 1936, T. Casperson demonstrated the ability to quantify chromosomal and cytoplasmic areas of the cell using different UV wavelengths [3]. These exciting experiments were carried out using the ultraviolet absorption measurement of a grasshopper metaphase chromosome. Casperson was able to show the differences between chromosomal absorption and cytoplasmic absorption. Almost 30 years later, Casperson’s son O. Casperson used similar techniques to show a comparison of DNA distribution in normal cervical cells, cervical carcinoma cells, and pre-malignant cells [4]. This work clearly demonstrated that both detection and quantification of single-cell properties could be a potential diagnostic tool by showing the frequency distribution of DNA from single cells.

However, some of the earliest cytometry with actual quantification was performed in 1951 by Robert Mellors, who designed a semi-automated instrument called a microfluorometric scanner using ultraviolet light with an objective lens to scan cells in order to measure extinction, and the cells were scanned with a cathode ray tube using a flying spot scanner. Mellors introduced the term “fluorochrome” and also identified the use of intrinsic fluorescence (or what we now call autofluorescence), as well as utilizing fluorescent dyes such as berbine. He was also an early user of a photomultiplier tube to collect fluorescence signals. His aim was to automate the classification of cytological criteria previously introduced by Papanicolaou [5,6]. A breakthrough in quantification was achieved by Mendelsohn in 1958 when he published a lookup table for calculating the effects of the distributional errors of Beers law when measuring the two-wavelength photometry of chromosomes [7].

Regarding cell suspension detection, Moldavan developed a simple photoelectric technique for counting cells in a liquid [8]. The goal of this approach was to accurately determine the number of bacteria in a liquid sample. By performing accurate plate counts, Moldavan was able to determine a relationship between his electronic counts and actual particle numbers. Moldavan’s aim was to quantify blood cells. However, due to the extremely weak signal strength obtained with commercially available electronics, this proved to be challenging. Nonetheless, this publication marks the first instance of successfully counting cells in a tube. Several patents were filed in the 1950s for developing blood cell counting technologies. These include Wallace Coulter’s 1951 submission for his Coulter Counter issued in 1957 [9], and Parker and Horst’s patent filed in 1953 to create a two-color blood cell detection device [10]. Coulter’s groundbreaking contribution to the field of cell counting came through his revolutionary device, the Coulter Counter, which he introduced in his sole published work [11].

In the realm of fluidic-based cell detection, the introduction of hydrodynamic focusing was a critical innovation. This principle, also known as sheath flow, was first described in Crosland-Taylor’s 1953 paper. Its significance lies in the fact that it serves as the foundation for virtually all flow cytometers. Therefore, Crosland-Taylor’s work was a seminal contribution to the field of cytometry [12].

## 2. Fluorescence Detection Principles

The principle of fluorescence has been around for over 100 years and is based on the ability to excite specific molecules to higher energy levels, which upon return to the ground state emit fluorescence at a longer wavelength but lower energy. Built upon the principle established by George Stokes [13] and defined as changes in energy levels described by Aleksander Jabłoński in 1933 [14], flow cytometry takes advantage of the Stokes shift of fluorochromes to collect a very sensitive signal well separated from the excitation. In such a case, multiple fluorochromes can be used simultaneously, allowing for high complexity analysis of cell populations. Typically, there are a number of common fluorochromes used in flow cytometry, as shown in Table 1. There are now an enormous number of “proprietary” dyes on the market—almost every major company has its own “house brand” of dyes, and it is beyond the scope of this review to identify them all.

Because it is possible to use multiple excitation sources, many dyes can be used in flow cytometry assays. As explained in a later section, how many of these and other dyes can be used in a single sample depends on the type of flow cytometer.

## 3. The Principles of Flow Cytometry

Flow cytometry originates from the Greek words “kytos,” meaning cell or container, and “metron,” meaning measure. Essentially, flow cytometry refers to the measurement of cells or particles in flow. Flow cytometers can only use cells in suspension. It is possible to disassociate tissue, as demonstrated by Hedley [15], and then run the single cells through a cytometer; however, this requires careful management of the cell suspension to avoid clumps, although it does demonstrate the potential for tissue analysis. The process of hydrodynamic focusing, as noted earlier [12], allows cells to be focused into a single line with the cells clearly separated from each other. In this case, each cell can be analyzed independently from any other cell, even when operating at tens of thousands of cells per second.

A flow cytometer is composed of several key components: a fluidic system that controls the flow of cells or particles through the instrument; an optical system composed of one or multiple lasers for excitation, multiple filters, mirrors, and detectors to capture the emitted fluorescence and scatter signals; and an electronics and data acquisition system that is usually integrated within software for data analysis, as shown in Figure 1.

Cell sorting is the physical separation of cells based on various properties that can be measured. Over many years, the sorting capacity of instruments has evolved into very high speed sorting primarily for the isolation of rare cells but also for very routine applications, such as sperm sorting. These advancements have enabled more precise and efficient cell detection and sorting, facilitating a wide range of applications in research, diagnostics, and therapeutic development. Cell sorting is the preferred method for single-cell recovery, particularly when needing pure populations, and, if necessary, delivered under sterile conditions. Cell sorting enables the recovery of specific cell populations for downstream applications such as cell culture, genomic analysis, and further functional studies. As shown in Figure 1**,** cell sorters can separate multiple populations of cells simultaneously. Here, four sorted populations are shown, but it is possible to sort six or eight populations depending on the instrument available. When sorting cells, it is critical to ensure that the medium in which cells are deposited is compatible with the metabolic support of the sorted population.

Regardless of all the advancements in flow cytometry since its inception, the basic principles have remained essentially unchanged for over 60 years, and the technique has primarily focused on fluorescence-based analysis. Flow cytometry traditionally relies on detecting fluorescence signals emitted by fluorochrome-labeled probes in order to analyze cellular properties and identify specific cell populations. This fluorescence-based approach has been fundamental in characterizing various biomolecules, including antibodies, nucleic acids, and intracellular markers, providing valuable insights into cellular function, differentiation, and disease states.

While fluorescence-based detection has been the cornerstone of flow cytometry, recent developments have expanded the technique’s capabilities. For instance, the introduction of spectral flow cytometry has enabled the simultaneous detection of a broader range of fluorochromes, addressing the limitations imposed by the spectral overlap [16,17,18]. In addition, the development of mass spectrometry technology has brought about the emergence of mass cytometry, also known as CyTOF. This innovative approach enables cells to be labeled with metal isotopes instead of fluorochromes, leading to the measurement of a significantly higher number of parameters [19].

Indeed, flow cytometry (“measurement of cells”) is not limited to fluorescence alone. Researchers have explored alternative detection methods, such as surface plasmon resonance (SPR) [20], surface-enhanced Raman scattering (SERS) [21,22], and electrical impedance [23,24,25], to complement or augment fluorescence-based analysis. These hybrid detection approaches aim to expand the range of analytes that can be measured and offer additional insights into cellular properties beyond fluorescence signals alone.

It is important to distinguish between analysis and sorting techniques because some analysis methods cannot be directly combined with sorting based on specific properties of the cells, fixation methods, employed labels, or analysis techniques. Instrument constraints, time limitations, and the potential for sample damage contribute to this distinction. As a result, sorting methods primarily rely on fluorescence-based detection because of the sensitivity of fluorescence and the vast array of fluorescent-conjugated antibodies that are commercially available.

In contrast, SPR methods, although non-destructive to samples, currently lack instruments that allow for the sorting of cells based on specific properties of interest. SPR is primarily used for label-free analysis of cellular interactions and molecular binding events.

With CyTOF methods, the atomization and ionization of cells during analysis make it currently infeasible to recover living cells after the analysis process; however, the relatively high parameter space and lack of needing to deal with spectral overlap since fluorescence is not used are clear advantages in mass cytometry.

In this review, we aim to discuss the aforementioned advancements and future developments in flow cytometry. We will delve into the improvements made in fluidic systems, including miniaturization and the integration of microfluidic technologies. Additionally, we will explore the enhancements in detection systems, optics, and software that have paved the way for more sophisticated and precise analyses.

Furthermore, we will examine the advancements in fluorescence dyes, focusing on those that exhibit improved stability, reduced spectral overlap, and increased quantum yield. These developments have expanded the range of detectable parameters and enabled more comprehensive and multiplexed analysis of cellular characteristics.

The utilization of fluorescence-coded beads and artificial compartments, such as emulsion systems and droplet microfluidics, will be discussed in relation to their ability to expand the applications of flow cytometry to secreted molecules, cytokines, and enzymes. We will explore the advantages and challenges associated with these novel approaches, highlighting their potential to advance research in various fields.

Moreover, we will mention the ongoing developments in data analysis techniques, including advanced algorithms, machine learning, and high-dimensional analysis methods.

Lastly, we will explore future directions and potential developments in flow cytometry. This will encompass emerging technologies such as spectral flow cytometry and the integration of flow cytometry with other complementary omics techniques. We will discuss the implications of these advancements in expanding the capabilities of flow cytometry and their potential impact on understanding cellular dynamics, disease mechanisms, and therapeutic interventions.

Through this comprehensive review, we aim to provide an overview of the recent advancements, current challenges, and future directions in flow cytometry, highlighting its potential to drive further breakthroughs in biomedical research, diagnostics, and personalized medicine.

## 4. Polychromatic vs. Spectral Cytometry

There are currently two forms of fluorescence-based flow cytometry in use. One has been the mainstay of the field for the past 45 years and is generally termed polychromatic flow cytometry. It is based on the principle of employing a dedicated sensor for each dye. The alternative is spectral flow cytometry, first implemented practically in 2004 [16,17,18,26,27,28], which uses a large bank of detectors to collect all of the fluorescence signals across the entire spectrum (Figure 2). Spectral cytometry has been more recently reviewed [28,29,30]. Spectral flow cytometry offers distinct advantages by utilizing multiple sensors to capture the complete fluorescence signals from all dyes, without the need to isolate each label’s dominant spectral band by an individual detector [31,32]. This approach involves employing a greater number of detectors compared to the number of utilized dyes, resulting in multiple representations of each label within the dataset.

The technical challenge of polychromatic cytometry lies in the overlapping signals of dyes with broad spectra and the difficulty in separating these signals [33]. The sensor output is inaccurate in reporting the actual signal from a fluor, resulting in incorrect reporting of the abundance of the labeled marker. The only solution is to compensate for the spectral overlap through mathematical estimation of the actual values. Compensation procedures are frequently carried out manually rather than utilizing automated algorithms, leading to inconsistencies. This creates difficulties when attempting to reproduce the same assay on separate occasions with complete accuracy. While spectral flow cytometry still faces issues with spectral overlap due to the complexity of spectral unmixing, it is always performed in an automated and, consequently, reproducible fashion. Paradoxically, this enhances the robustness of the data analysis and ensures that the assays are performed consistently. Typically, the unmixing process is assisted by a database containing each dye’s unique spectrum, allowing the determination of the component parts of the mixed signal.

## 5. Applications of Flow Cytometry

Before delving into the numerous applications of flow cytometry, it is essential to emphasize the distinction between analysis and sorting capabilities. In terms of analysis, flow cytometry enables the recording of many readouts for each individual cell and across multiple samples. On the other hand, cell sorting involves the specific isolation and recovery of the desired cell population from the rest.

Flow cytometry is a versatile technology with a wide range of applications. It is commonly used for immunophenotyping, which involves identifying cells based on their surface or intracellular markers, as well as for viability assays and cell cycle analysis. However, flow cytometry can also be used for more specialized applications, such as studying ion fluxes, analyzing cytokines, and engineering proteins.

Flow cytometry’s remarkable strength lies in its capacity to analyze vast populations of individual cells, rendering it an invaluable tool for unraveling cellular heterogeneity and identifying rare cell subsets. This capability to examine cells at a single-cell level empowers researchers to discern and characterize distinct subpopulations, shedding light on the intricacies of cellular diversity and elucidating the significance of rare cellular events within disease processes or developmental pathways. Furthermore, flow cytometry sorter instruments provide the added functionality of physically sorting cells from any subgroup, expanding the realm of possibilities in cell isolation and downstream applications.

In the following sections, we will explore these applications in detail, highlighting the significance of flow cytometry in advancing our understanding of cellular biology, disease mechanisms, and therapeutic interventions. We will also discuss emerging trends, technological advancements, and future prospects that further enhance the potential of flow cytometry in various research areas. Through this exploration, we aim to underscore the versatility and impact of flow cytometry as a powerful tool for cell analysis, sorting, and gaining deeper insights into complex biological systems (Figure 3).

### 5.1. Cell Phenotyping

Immunophenotyping is a prominent application of flow cytometry that involves the identification and characterization of immune cell populations based on their surface or intracellular markers.

Flow cytometry enables the precise identification and classification of immune cell populations, including T cells, B cells, natural killer (NK) cells, dendritic cells, monocytes, macrophages, platelets, and granulocytes. By utilizing specific combinations of fluorescent-labeled antibodies targeting unique cell markers, researchers can discern and quantify various immune cell subsets within a heterogeneous population. By analyzing the intensity of fluorescence associated with specific markers or in spectral cytometry and acquiring the complex spectroscopic signatures of any cell, we can evaluate changes in marker expression during immune cell activation, differentiation, or disease progression. This profiling helps in understanding the functional properties and phenotypic characteristics of immune cell subsets. This is instrumental in diagnosing and monitoring various hematological disorders, such as leukemia and lymphoma. By analyzing specific surface markers or abnormal antigen expression patterns on malignant cells, flow cytometry assists in subtype classification and disease monitoring to guide appropriate treatment strategies.

Immunophenotyping is crucial in monitoring immune responses in research studies as well as in clinical trials. It enables the assessment of immune cell activation, proliferation, and functional changes in response to stimuli, vaccines, or therapeutics. This information contributes to understanding immune mechanisms, evaluating treatment efficacy, and developing immunotherapeutic strategies. With the advent of monoclonal antibodies [34] and the subsequent development of specific T cell clones by Schlossman [35] in conjunction with Ortho Diagnostics, commercial monoclonal antibodies demonstrated perfect integration into flow cytometry. Subsequently, the new approaches to flow cytometry instrumentation by Herzenberg [36,37,38,39], the subsequent development of 2-color fluorescence (using polyclonal antibodies) [40], and eventually, the identification and sorting of conjugated monoclonal antibodies [41] facilitated the development of the vast numbers of monoclonals available today.

The application of flow cytometry extends beyond immunophenotyping to the phenotyping and characterization of various non-immune cell populations. It plays a crucial role in identifying and isolating stem cell populations, such as hematopoietic stem and progenitor cells, based on specific surface markers like CD34 and CD133 [42,43,44,45,46]. Additionally, flow cytometry facilitates the characterization of cancer cells, including cancer stem cells, by staining them with specific markers to assess antigen expression and investigate therapeutic targets and resistance mechanisms. In neurobiology, flow cytometry is employed to analyze specific neuronal subtypes, glial cells, and neural progenitor cells using surface markers like CD133, CD56, and neuronal markers. Moreover, flow cytometry aids in the phenotypic analysis of endothelial cells involved in vascular development, angiogenesis, and vascular-related diseases. By examining surface markers, researchers can identify endothelial cell populations, investigate their function, and study their activation in disease states. Flow cytometry also enables the phenotypic characterization of mesenchymal cell subsets, including mesenchymal stem cells and fibroblasts, by utilizing markers such as CD90, CD73, and CD105. Additionally, it can be applied to analyze epithelial cell populations, identify and characterize different subtypes, study the epithelial–mesenchymal transition (EMT) [20], and investigate cellular dynamics in tissue development and disease progression.

Flow cytometry has also emerged as a valuable tool for the phenotyping of microbial cells, enabling the rapid analysis and characterization of various microbial populations [47,48,49,50]. It can be used to identify and classify different bacterial species based on their phenotypic characteristics. By using fluorescent dyes or antibodies targeting specific microbial markers, researchers can estimate the abundance of different microbial species or cell types present in a sample, providing insights into population dynamics and microbial community composition.

### 5.2. Viability Assays

Flow cytometry enables the quantitative measurement of viable and non-viable cells within a population, as first shown by Jovin’s group [51]. By using fluorescent dyes that selectively label either live or dead cells, flow cytometry can provide precise and reliable viability measurements, facilitating the determination of cell viability percentages. In addition, flow cytometry offers high sensitivity for viability assays, capable of detecting even subtle changes in cell viability [52,53,54,55,56]. It can discriminate between viable, apoptotic, and necrotic cells based on specific markers or dyes, providing a more detailed understanding of cell health and status. It also allows for multiparameter analysis, simultaneously assessing cell viability and other cellular characteristics. By combining viability dyes with markers for cell surface antigens, intracellular proteins, or functional assays, researchers can obtain comprehensive information about cell viability in the context of specific cell types or experimental conditions. Different cell types may exhibit variations in their responses to viability dyes or markers due to inherent biological differences. Flow cytometry enables researchers to tailor viability assays to specific cell types of interest, enhancing the accuracy and relevance of the results. By repeatedly analyzing samples at defined time intervals, researchers can track viability alterations in response to various stimuli, treatments, or environmental conditions, providing valuable insights into cell behavior and response kinetics. As assessments are performed at the single-cell level, flow cytometry provides information on the viability status of individual cells within a population. This capability is particularly valuable when studying heterogeneous cell populations or investigating rare cellular events, enabling researchers to identify and analyze subpopulations with distinct viability characteristics.

Viability assessment using flow cytometry and other methods relies on the use of various small-molecule dyes with different hydrophobic properties, some of which can penetrate intact cellular membranes while others cannot. The most commonly used dyes for viability detection, such as propidium iodide and 7-aminoactinomycin D, bind to DNA but can only enter cells with compromised membranes, rendering dead cells fluorescent. Although these dyes remain widely used, newer fluorescent molecules with diverse spectral properties have been developed to enable better multiplexing with other assays while still adhering to the same underlying principles.

An alternative approach utilizes annexin V, a protein with strong binding affinity for phosphatidylserine, which becomes exposed on the outer surface of the plasma membrane during the early stages of apoptosis [57,58,59]. In flow cytometry, fluorescent-labeled annexin V is combined with viability dyes to differentiate viable cells (annexin V-negative) from apoptotic cells (annexin V-positive). This assay, akin to antibody-based techniques, employs fluorescent-labeled annexin V as a probe for detecting apoptotic cells. The versatility of annexin V conjugation with various fluorescent dyes provides a wide range of color options for fluorescence detection in this assay. The annexin V/PI assay is widely used to differentiate between apoptotic and necrotic cells. In this assay, cells are simultaneously labeled with annexin V and propidium iodide.

There is considerable variability in antigen abundance after fixation. Some markers exhibit almost normal presence when cells are phenotyped after fixation [60]; in some cases, the total intensity of each marker is reduced while the autofluorescence is increased [61]. On the other hand, several studies have shown that the expression of antigens is either decreased or completely lost after fixation [62]. Clearly, if cells are to be fixed and stored prior to analysis, it is critical to check each and every antigen of interest pre- and post-fixation.

Lastly, viability studies also employ cell-permeant fluorescent dyes that can enter live cells but only become fluorescent upon interaction with intracellular enzymes [63,64,65]. Examples include calcein AM, carboxyfluorescein diacetate (CFDA), and fluorescein diacetate (FDA). These dyes are taken up by viable cells and converted by intracellular esterases into their fluorescent forms. As enzymatic activity is crucial, the choice of available dyes for this type of assay is limited, typically utilizing the fluorescein channel for detection. By utilizing these dyes and assays, viability assessments using flow cytometry enable the discrimination of live, apoptotic, and necrotic cell populations, providing valuable information about cell health and integrity.

### 5.3. Cell Cycle Analysis

From the earliest days of flow cytometry, cell cycle analysis became a valuable application [66,67,68]. The key is the relationship between some specific fluors and the amount of nucleic acid. Several dyes have been identified as binding nucleic acids, such as propidium iodide (PI), Hoechst, DAPI, 7-aminoactinomycin D, mithramycin, ethidium bromide, and others. Some dyes have more specificity to RNA, such as thiazole orange and thioflavin. PI is also known as a stoichiometric dye that intercalates between bases. Because of this, there is a stoichiometric relationship between the amount of dye and the amount of DNA. Because of this property, PI can be used to accurately quantify the amount of DNA, and when carefully run through a flow cytometer, even small changes in nucleic acid content can be easily identified.

There are a significant number of approaches for combining flow cytometry with cell cycle analysis. For example, the DNA/Ki67 assay can combine phenotype selection with cell cycle analysis [69] for monitoring p53 cell cycle arrest [70], evaluating anticancer activity [71] and sperm cell fractionation [72], and determining multidrug resistance [73], among many other application. It is highly recommended to refer to Ligasova’s review for a comprehensive analysis of this subject [74].

### 5.4. Ion Flux Assays

Calcium, acting as a critical secondary messenger, plays a vital role in numerous cellular signaling pathways. It is particularly important in immune cell activation, including T cells [75], B cells [76], and natural killer (NK) cells [77]. Additionally, calcium signaling is involved in mast cell degranulation, a crucial process in allergic reactions and immune responses [78]. Moreover, calcium flux is essential for neuronal excitability, synaptic transmission, and neurotransmitter release [79]. Given its significance, the study of calcium flux holds immense importance in various medical and drug discovery applications.

Flow cytometry, with its ability to analyze individual cells within a heterogeneous population, offers a high-resolution assessment of calcium flux dynamics at the single-cell level. This empowers researchers to identify subpopulations exhibiting distinct calcium signaling patterns and to explore cell-to-cell variability within a sample. Furthermore, flow cytometry allows for multiparametric analysis by integrating calcium flux assessment with other markers or functional assays.

While monitoring calcium flux via flow cytometry requires specific tools, it offers unique advantages. Early measurements of cellular degranulation were determined by flow cytometry using calcium ionophore A23187 [80]. Valet also measured intracellular calcium by flow cytometry [81]. Fluorescent dyes such as fluo-3 [82,83] and indo-1 are commonly used for calcium flux determination by flow cytometry [84,85].

In mammalian cells, alterations in intracellular calcium concentration are among the most rapid responses to various stimuli, sometimes occurring within nanoseconds. However, recording these rapid calcium responses poses a challenge due to gaps in the data caused by the addition of compounds, resulting in the loss of detailed information. To address this, modifications have been made to flow cytometers to enable continuous measurement while compounds are added to the sample [86].

Although Ca^2+^ flux measurement is one of the most common applications, other ions like magnesium [87], potassium, sodium [88], and hydrogen [89] can also be monitored using similar techniques.

### 5.5. Cellular Function Measurement

Because flow cytometry analyzes single cells, it is possible to obtain very high-quality data on an entire population of cells from a functional perspective. Some of the earliest measurements were of cellular esterases [90]. For example, a number of studies defined the oxidative potential of granulocytes using reactive dyes that respond to changes in oxidation state. For example, hydroethidine was used to demonstrate neutrophil respiratory burst [90]; similarly, monocyte functions have been demonstrated by flow cytometry [90]. Other probes, such as dichlorofluorescein diacetate, have been used for phagocyte function studies [90,91,92,93]. Some studies on cell function could only be performed by flow cytometry because of the very few cells available, such as crevicular neutrophils [94]. Studies on phagocytosis by flow cytometry were begun in the early 1980s [95,96,97,98,99], and a variety of approaches have been developed since [100,101,102,103].

### 5.6. Protein Engineering

Flow cytometry and sorting have not traditionally been among the most common techniques used in protein engineering. However, in recent years, there has been increasing utilization of these techniques in the field. Several reports have highlighted the usage of florescence-activated cell sorting (FACS) systems for protein evolution studies on enzymes such as cytochrome P450 [104], glucose oxidase [105], chitinases [106], cellulases [107,108], peroxidases [109,110], esterases [111], transferases [112], beta galactosidases [113], thiolactonases [114], and a few other enzymes.

Protein engineering has emerged as a powerful tool for tailoring proteins to meet specific functional requirements. Conventionally, directed evolution approaches have been employed, involving the introduction of mutations (random or specific) at the gene level to create libraries consisting of thousands to millions of individual protein variants (Figure 4A). The screening of these libraries for desired properties typically relies on colony picking and microtiter plate assays. However, this process is slow, expensive, generates a substantial amount of plastic waste, and requires a significant amount of consumables. Thus, coupling the screening process with a high-throughput technique such as FACS offers significant advantages. FACS enables the analysis of up 10^8^–10^9^ clones per day and the sorting of clones exhibiting the desired properties.

One limitation of applying flow cytometry in protein engineering is the need to maintain the connection between the genotype and the phenotype throughout the entire process. In the classical screening approach, this connection is preserved within individual wells of a microtiter plate, where a single cell expressing a single mutant protein is assayed. In flow cytometry, new methods must be employed to ensure this connection. One strategy involves designing a fluorescence assay in which the product of the enzymatic reaction being improved either covalently labels the cells [110,115,116] or remains trapped within the cells producing the respective mutant enzyme [111,112,117,118]. However, this approach has limitations, as developing an assay with the desired properties that accurately reflects the properties of the enzyme being improved can be challenging in some instances.

An alternative approach involves the creation of artificial compartments surrounding individual cell clones during the assay (Figure 4B). Typically, these compartments take the form of water-in-oil-in-water double emulsions since the sheath fluid of the flow cytometer consists of water-based buffers, and the compartments need to be compatible with the aqueous phase. While a few successful applications of this approach have been reported [106,107,113,114,119], the emulsions generated using conventional methods tend to be highly heterogeneous. As a result, there is a growing trend towards adopting microfluidic devices as they offer more uniform emulsion and compartmentalization protocols. Additionally, since analysis and sorting can also be performed on microfluidic chips, there has been a tendency lately to move away from using flow cytometers for these tasks.

### 5.7. Bacterial Cell Sorting

The application of flow sorters for bacterial isolation and detection is fascinating, as rapid detection and sorting of single bacterial cells in a suspension is possible [120,121] compared to time-consuming conventional agar plating-based detection methods. Bacterial cells can be sorted at high throughput and collected in the desired outlet for further analysis. Despite the high-performance capabilities of cell sorters, their application in microbiology has been limited. This is mainly due to the small size of microorganisms, which makes it difficult to differentiate them from cell debris or background particles in the media [122], and also in some cases due to pathogenicity. In addition, the similarity in size and shape among different subpopulations of bacteria makes it highly challenging to sort them efficiently [123]. Another major problem is linked to the concentration of bacteria in the sample. Gating the population of interest is difficult, which leads to inconsistent sorting when the concentration of bacteria is very low [121]. If the starting sample is not pure or there is a need to separate sub-populations of organisms, multiple sorting and enrichment steps may be needed, which will further affect cell recovery [121]. Even though whole blood can be directly used, usually red blood cells are lysed before introduction to the cell sorter [124]. However, more efficient sample preparation steps may be needed if low concentrations of bacteria are to be detected in blood [125,126]. By combining staining and scatter data, Irene et al. in 2007 showed that it was possible to detect bacteria as low as 1 CFU/mL in cell culture production medium [127]. However, high concentrations of bacterial protein or DNA were needed for downstream analysis. The recommended concentration of sorted cells for protein analysis is around 10^9^ cells/mL [122] and about 10^3^ or 10^4^ cells for nucleic acid analysis [128]. Another potential issue is that antibodies that are specific to bacterial strains are often not available, which limits detection to a few bacterial strains. Nowadays, proteomic strategies are gaining interest for detecting biomarkers that are specific to bacterial species and strains [129,130].

Other factors that restrict the applicability of cell sorters for bacterial detection and sorting are mostly related to the sorter hardware capabilities themselves. In the early days of flow cytometry instruments, a limited number of lasers and detectors restricted the use of only one or two types of fluorescent dyes at a time [131]. With the development of more recent instruments, the number of lasers and detectors has increased, allowing multiplexing capabilities [18]. Some of the examples of modern cell sorters with multiplexing include: Thermo Fisher’s Bigfoot spectral cell sorter (9 lasers and up to 64 detectors), BD FACSAria III sorter (6 lasers and 20 detectors), Sony MA900 Cell Sorter (4 lasers and 14 detectors), and Beckman Coulter’s MoFlo Astrios EQ (7 lasers and up to 44 detectors), among others [132]. Regardless of these developments, sorting bacteria for clinical applications is not particularly easy, especially if the bacteria are pathogenic and can cause health risks to the users. This demands proper management of samples and waste.

Conventionally, pathogenic bacteria are handled inside a biosafety level 2 (BSL 2) hood, which ensures safe handling. Operating and sorting pathogenic organisms using flow cytometers requires maintaining the machine inside a BSL 2 hood [133]. The problem is that many sorters do not fit inside a BSL 2 hood, and handling flow sorters inside the hood can be exhausting to the user and makes service complicated. This has prevented the use of flow sorters for clinically relevant pathogens. One way to solve this problem is by incorporating biosafety features into flow sorters. The use of regular Class 2 hoods with HEPA filters was used with cell sorters from BD Biosciences, such as FACStar and FACSVantage, using FACSDiVa software [133]. Recently, Thermo Fisher’s Bigfoot spectral cell sorter was released into the market with advanced safety features [134]. Bigfoot is a spectral cell sorter that uses the spectral signatures of the dyes for real-time unmixing and sorting of cells of interest [134]. It is equipped with special biosafety features similar to a BSL2 hood with a HEPA filter for aerosol and biocontamination management. Six samples can be maintained simultaneously in a separate sample chamber that limits exposure of the sample to the user when running the machine. A separate sorting chamber is provided in which sorting can be performed only when the door is closed, preventing splashing of the sample on the user. These features facilitate the safe handling of pathogens, and the Bigfoot sorter can be therefore operated within standard laboratory settings. In addition, the Bigfoot cell sorter has the capability of handling up to 100,000 cells/second in multi-stream sorting mode, demonstrating high-throughput abilities [134]. The combination of spectral sorting and biosafety features can further enable the handling of pathogenic organisms with safety.

Since the sorting of pathogens is relatively recent, we have included some material in this review from work within our laboratory. To demonstrate bacterial sorting using the Bigfoot cell sorter, we sorted and deposited bacteria at desired spots on agar plates. Using the 96-well plate map available in Bigfoot software (Figure 5A), we deposited single non-pathogenic *Salmonella enteritis* cells on Xylose Lysine Tergitol-4 (XLT4) agar and Salmonella Shigella (S.S) agar. Both of these agars are selective agars that permit the growth of *Salmonella* species [135,136]. The agar plates were incubated at 37 °C for 24 h. Figure 5B shows the colonies formed at the deposited spots on both agar plates. It can be observed that some spots did not show colony formation, which could be attributed mainly to the mathematical limitations of Poisson’s distribution, which reflects the probability of a droplet encapsulating a cell [137]. This rapid, precise, and controlled deposition of cells at the desired spots opens up new possibilities for tracking the growth of a single bacterial cell as it develops into a colony. In addition, the biosafety features in the Bigfoot sorter enabled the handling of pathogenic organisms without needing special facilities. It is important to note that more optimization is needed to establish a safety protocol and prevent contamination before handling pathogenic organisms using Bigfoot.

### 5.8. Droplet Microfluidics

Droplet microfluidics is a relatively new field that focuses on the formation, manipulation, and analysis of discrete droplets containing cells or DNA in picoliter volumes. T-junctions [138,139,140] and flow focusing [141,142,143] are the most common approaches employed in microfluidic systems to create emulsion compartments, each having specific advantages [144] in terms of droplet size control, throughput, and functionality [145]. These techniques enable the precise formation of water-in-oil or oil-in-water droplets. Once formed, the droplets can be manipulated and processed within the microfluidic system through operations like merging [146,147,148], splitting [149,150,151], analysis, and sorting [152,153].

Droplet microfluidics offers several advantages over traditional bulk-scale methods. Firstly, it enables the handling of small volumes of reagents, reducing consumption and waste. Secondly, the compartmentalization of reactions into individual droplets allows for high-throughput analysis and parallelization of experiments, enabling rapid screening and optimization of reaction conditions. Additionally, droplet microfluidics provides excellent control over reaction kinetics, facilitating precise timing and monitoring of reactions. The small droplet sizes also promote efficient mixing and heat transfer, leading to enhanced reaction performance and reduced reaction times. The applications of droplet microfluidics span a wide range of fields, including biology, chemistry [154,155], material science [156,157], and medicine [158,159]. In biology, droplet microfluidics enables single-cell analysis, high-throughput screening of biomolecules, studies of cellular heterogeneity, and drug discovery.

Flow cytometry analysis is a powerful technique for studying single cells, providing valuable information about various parameters. However, its measurements are limited to molecules that are directly connected to the cells, such as surface or intracellular markers. This limitation restricts the ability to study molecules that are secreted by cells or produced by DNA molecules but are not physically attached to them.

Droplet microfluidics, on the other hand, offers a novel approach to overcome this limitation. Encapsulating cells or DNA within individual droplets creates discrete compartments that enable the analysis of compounds that are released or generated by the encapsulated entities. The droplets act as miniaturized reaction vessels, preserving the connection between the molecules of interest and enabling their characterization even when they are not directly connected to the cells or DNA.

Additionally, the application of droplet microfluidics expands to DNA analysis [160,161]. Encapsulating DNA molecules within droplets enables the detection and quantification of nucleic acids as well as the characterization of enzymatic activities or gene expression profiles.

Some valuable applications of droplet microfluidics that are not yet widely used include the selection of stable cell lines and high protein producers for industrial manufacturing purposes, offering a more efficient and rapid alternative to traditional methods such as microtiter screening and cloning by limiting dilution.

Generating a vast number of droplets containing single cells creates a high-throughput screening platform. Each droplet acts as an isolated microscale bioreactor [162,163], providing a controlled environment for cell growth and selection. This setup enables the parallel evaluation of numerous clones, significantly reducing the time and effort required to identify the desired traits, such as stable production of a specific product or resistance to specific conditions.

Similarly, droplet microfluidics has proven to be highly beneficial in the field of protein engineering [164,165]. In protein engineering, the generation and screening of large gene libraries containing millions of variants is a crucial step to identify improved protein properties such as enhanced stability, altered specificity, or increased activity. Traditional methods of library screening are time consuming and labor intensive, as previously discussed.

By leveraging droplet microfluidics, entire gene libraries can be encapsulated within droplets, with each droplet containing a unique gene variant. This allows for the parallel screening of many variants within a single experiment. The droplets act as isolated reaction compartments, facilitating the expression of encoded proteins and subsequent functional assays. This approach dramatically speeds up the screening process, enabling the evaluation of tens of millions of protein variants in a single day.

Droplet microfluidics has proven to be successful in directed evolution campaigns for screening a range of enzymes, including cellulases [166], glucose oxidase [167], aldolases [168], xylanases [169] DNA polymerases [170], transaminases [171], and a few others.

Most of the assays reported are based on fluorescence-assisted droplet sorting (FADS). In the last five to ten years, a couple of new detection methods have been adapted for droplet microfluidics analysis and sorting. These include detection based on absorbance [172], mass spectrometry [171,173], electrochemistry [174], Raman spectroscopy [175], and even NMR [176]. The advantages and limitations of these methods have been reviewed elsewhere [164]. Overall, the integration of droplet microfluidics with advanced analytical techniques, such as the ones described above, offers unprecedented capabilities in protein engineering and single-cell characterization. One additional advantage of microfluidics systems over classical flow cytometers is the utilization of single-use fluidics. Chips can be designed as disposable devices, eliminating the need for cleaning and reducing the risk of contamination, thus enabling the use and screening of BSL-3 and BSL-4 pathogens.

Droplet microfluidics, despite its numerous advantages, is not without its drawbacks. One notable limitation is the analysis and droplet generation speed, which typically ranges from 1000 to 4000 Hz and can sort up to 1000 events per second. In comparison, classical flow cytometry instruments can achieve much higher analysis speeds, reaching up to 70,000 events per second.

An effective strategy involves the integration of microfluidic technology for generating double emulsions, which are subsequently analyzed and sorted using conventional flow cytometers and sorters [177,178]. This approach combines the advantages of microfluidics in creating uniform compartments with the rapid processing capabilities and detection methods provided by flow cytometry systems.

Furthermore, the availability of instruments designed specifically for droplet microfluidics is another limitation. While droplet microfluidics has gained significant attention and interest, the number of commercially available instruments specifically optimized for droplet-based analysis is relatively limited. Even within research laboratories, most available instruments are often designed for basic functionalities such as one or two fluorescence detection or sequencing capabilities. This limited availability of specialized instruments can restrict the scalability and accessibility of droplet microfluidics for researchers in specific applications.

Nevertheless, it is essential to note that ongoing advancements in droplet microfluidics are continually addressing these issues in order to overcome current limitations.

The adaptation of detection systems designed for flow cytometers into microfluidic chips promises to unlock the full potential of droplet microfluidics, making it an unstoppable and indispensable technique in biology research labs. It enables enhanced analysis, improved throughput, and expanded applications, empowering researchers to delve deeper into the complexities of cellular systems and accelerate scientific discoveries.

### 5.9. Next-Generation Bioengineered Biologics

The bioengineered drug market, excluding COVID-19 vaccines, had a value of $327 billion in 2021 and is projected to reach $521 billion by 2027 [179]. This growth is predominantly driven by therapeutic proteins, accounting for 65% of the market, followed by vaccines at 20% and other products at 15%. Monoclonal antibodies (mAbs) form a prominent product group due to their versatility in binding to a wide range of disease targets and their amenability to protein engineering modifications [180]. mAbs serve as reagents, diagnostics, and therapeutics, with OKT3, a murine CD3-specific transplant rejection drug, being the first FDA-approved therapeutic mAb in 1986. To date, over 130 monoclonal antibodies have received regulatory approval [181,182]. The pipeline for mAbs is robust, and advancements in protein engineering have facilitated the production of modified antibody molecules, including variations in size and the development of chimeric and humanized antibodies. The high specificity and low immunogenicity of antibodies have led to their widespread use in diagnosing and treating life-threatening diseases. The antibody market is driven by unmet medical needs, the rising incidence of various diseases, and the expanding global population. Notably, significant market opportunities exist in developing countries where such therapies are currently unaffordable. To address the challenge of manufacturing costs, faster identification of high-affinity target-specific antibodies as well as improvements in production methods are crucial.

State-of-the-art technologies such as single-cell isolation have revolutionized the discovery and development of antibodies, addressing the limitations of traditional methods [183]. Instead of relying on time-consuming and expensive processes involving the injection of disease targets into mice, isolation of B cells, and hybridoma generation and screening, cutting-edge techniques have emerged. Single B cell repertoire analysis and clonal expansion-guided identification via next-generation sequencing (NGS) have enabled the isolation of high-affinity neutralizing antibodies by cloning immunoglobulin genes directly from human survivors [184,185], eliminating the need for mouse immunization and subsequent humanization. By identifying and recruiting individuals who have recovered from infectious diseases like HIV, influenza, COVID-19, and Zika, human B cells can be directly isolated. These cells have undergone in vivo maturation, significantly increasing the likelihood of rapidly identifying high-affinity and neutralizing antibodies. This method, which does not require prior knowledge of antibody structure, leverages real-world patient and epidemiological data to identify and engage promising donors [186]. However, this approach is costly and time consuming, restricting analysis to only the most expanded clones, without considering rare clones that primarily reside in germinal centers, which may bind and neutralize the target antigen more efficiently. To overcome these challenges, a promising new strategy could involve coupling single-cell isolation with functional screening using flow cytometry, MACS, or microfluidics, thus reducing development costs and eliminating failed candidates.

Primary human B cell isolation from peripheral blood presents challenges in terms of sample availability, cell contamination, and low yields. Although methods like FACS, MACS, and microfluidics have been utilized to enhance purity and yield, they can negatively impact cell viability and functionality, requiring further improvement [187]. One potential solution lies in leveraging droplet microfluidics to optimize antibody discovery processes [188]. This can be achieved by employing standardized data-rich approaches and highly-integrated workflows, focusing on the development of novel functional assays while ensuring the preservation of B cell viability. Rather than utilizing the entire B cell population, the separation of pan B cells, memory B cells, and plasma cells using flow cytometry, MACS, or microfluidics based on surface markers (CD19, CD27, and CD38) should be considered [189]. These B cell subpopulations can then be stimulated with target antigens to activate and induce the proliferation of rare memory B cells, facilitating affinity maturation through activation-induced cytidine deaminase (AID) and somatic hypermutations (SHM), as described in [190,191,192]. Lanzavecchia’s work also revealed that T cell-produced IL-10 and IL-21 regulate the differentiation of B cells into antibody-secreting plasma cells.

The isolation of viable and functional B cells using FACS, MACS, or microfluidics poses challenges in antibody discovery workflows. To address these challenges, recent advancements in microfluidics have combined single-cell isolation with droplet-based FACS/microfluidics techniques, ensuring the preservation of correct heavy/light-chain pairings during functional screening and next-generation sequencing (NGS). Two automated platforms, the Berkeley Lights Beacon and the Sphere Fluidics Cyto-Mine systems, have been introduced to enable high-throughput screening of thousands to millions of B-cells within a short timeframe [193]. These platforms offer the ability to analyze and manipulate individual B cells [194,195] and can be seamlessly integrated with downstream workflows, such as cell culture, functional assays, and sequencing. Cyto-Mine employs a picodroplet-based platform, encapsulating single cells in picoliter-scale droplets containing culture medium, assay reagents, and fluorogenic substrates [195]. The system utilizes fluorescence-activated droplet sorting (FADS), in which droplets containing the cells of interest are sorted based on their fluorescence signals, enabling high-throughput sorting and recovery of selected cells within a microfluidic sorting chip [183].

Droplet microfluidics has become an established technology in supporting antibody discovery workflows, although continuous improvements are being reported. It involves the creation of water-in-oil emulsions using microfluidic techniques, generating numerous spatially separate reaction compartments at a small scale. Each droplet, containing suitable media to maintain B cell viability and assay reagents, can be seeded with a single B cell. As cells express antibodies, their detection is achieved through fluorescence binding assays. Droplets containing cells can then be sorted without damaging the cells by flowing them at high speed through additional microfluidic geometries. Droplets offer advantages such as spatial separation, localized retention of secreted molecules, and isolation of cells from high shear stress during rapid movement, making them highly relevant for antibody discovery. However, droplet microfluidics also have limitations. Currently, all B cell screening platforms are based on binding assays that allow only one or two parameters to be analyzed. By combining a microfluidic chip design, droplet microfluidics and multiparameter detection systems should enable multiparameter analysis on the same single cell. The current assessment of cell performance based on fluorescence measurements at a limited number of wavelengths necessitates further post-sorting assessment and quantification, leading to additional time required to identify the preferred cells in a population. Moreover, the current throughput and capacity of microfluidics systems are insufficient and too slow to assess an entire B cell repertoire in a single operation using conventional one-cell-at-a-time approaches. Enhancing the throughput from 4 × 10^7^ cells to 1 × 10^8^ cells is a key challenge, along with finding better strategies to explore the full repertoire of B cells within the droplet format, in order to achieve high precision and improve the overall specificity of the process of identifying best-in-class ‘developable’ antibodies through data-rich screening in a unified format ecosystem. Next-generation FACS, MACS, microfluidics, multiparameter analysis, automation, and process standardization will play a crucial role in achieving these objectives.

## 6. Flow Cytometry Data Science and Informatics

Sophisticated data processing informatics and machine learning techniques, all of which fall under the broad umbrella of artificial intelligence (AI), are at the heart of modern life sciences [196]. Recently, AI has captured substantial attention thanks to its impressive progress in various fields, such as medical image recognition and analysis [197]. In the context of cytometry, AI plays a crucial role in several areas, in particular, transforming data analysis and interpretation [198,199,200].

This review does not aim to offer a detailed analysis of the growing field of AI and data science in cytometry, given the numerous reviews already available on the subject [199,201,202,203,204,205]. However, we will highlight a few examples that demonstrate the significant current and future impact of data science on this area.

### 6.1. Spectral Unmixing and Compensation

As already mentioned, the advent of spectral cytometry introduced complexities beyond the traditional polychromatic approach [31,206,207]. While simple matrix inversion has been used for inter-channel compensation (unmixing) in traditional cytometry [33,208], spectral cytometry requires more advanced methods [27]. Although matrix pseudo-inverse, representing a closed-form solution to least-square minimization, seems like a natural choice for spectral unmixing, following work in remote sensing [209], the implicit assumption of Gaussian distribution for fluorescence measurement noise is fundamentally flawed due to stochastic photon emission and the nature of photodetection [32]. Therefore, alternative noise models, such as those accounting for Poissonian uncertainty or even overdispersed models, need to be considered. These physical limitations open up opportunities for creative data pre-processing approaches that incorporate various constraints into unmixing models [32,210]. In the future, we will likely see the use of data-driven noise models, which characterize the process of photon emission and collection via statistical rather than mechanistic representations and generate unmixed data employing the latest feature-learning techniques.

### 6.2. Data Pre-processing and Curation

Flow cytometry often involves the acquisition of large datasets from multiple samples, sometimes performed over an extended period or across multiple facilities. Furthermore, experiments may span a significant duration, resulting in varying data collection conditions. To address these challenges, pre-processing software is employed to perform tasks such as batch-to-batch normalization, outlier removal, and correction of other data acquisition errors that may arise during cytometry experiments [198,211,212,213,214,215]. These automated data pre-processing techniques ensure the quality and consistency of the data for subsequent analysis.

### 6.3. Data Visualization

The substantial amount of data generated by flow cytometry requires effective visualization techniques for interpretation and decision making. Traditional 2D dot plots displaying fluorescence intensities associated with markers of interest may be insufficient in high-dimensional cases. Therefore, cytometrists increasingly rely on visualization approaches that incorporate data reduction techniques. Spanning trees, self-organizing maps, and, most importantly, manifold learning models (including t-SNE, UMAP, PHATE, and others) are commonly utilized to reveal complex relationships and structures within multidimensional cytometry data, enabling a better understanding of cellular heterogeneity and identification of distinct cell populations [216,217,218,219,220].

### 6.4. Automated Gating and Cell Population Identification

Historically, flow cytometry data interpretation heavily relied on trained experts who manually gated and identified specific patterns of fluorescence co-occurrences associated with particular cell types. However, this process is labor intensive, subjective, and prone to inter-operator variability. Expectedly, the initial use of machine learning algorithms in cytometry was for gating purposes [221,222,223,224,225,226,227,228,229]. As a result, multiple approaches have been developed that directly or indirectly learn from expert-defined gates and consistently apply gating criteria across large datasets. This route improves efficiency, reduces human bias, and enhances reproducibility. Additionally, machine learning algorithms can be trained to identify and classify different cell populations based on their fluorescence profiles, even in the presence of overlapping populations or rare events. A plethora of methods can be employed for this task, from clustering and supervised learning through various Bayesian techniques and feature learning (deep learning) approaches [230,231,232,233,234,235]. These algorithms provide accurate and reliable cell population identification, facilitating a comprehensive exploration of cellular heterogeneity and, if used in clinical settings, an understanding of disease mechanisms [232,236,237,238,239].

### 6.5. Biomarker Discovery and Predictive Modeling

Statistical machine learning techniques have proven valuable in biomarker discovery when paired with various omics techniques [240,241,242]. By integrating flow cytometry profiles with other omics data, such as genomics, metabolomics, and proteomics, AI algorithms can improve the ability to identify novel biomarkers or combinations of markers associated with specific cell populations or disease states. Although this is still a relatively underappreciated direction in cytometry, the fundamental logic behind these techniques is quite simple and has been employed in other omics studies, such as lipidomics [243]. The approach involves feature selection procedures, which operate under the assumption that the features most essential for optimal classifier performance are likely to be associated with the mechanisms of the underlying biological processes [244,245]. This methodology employs classification models with embedded feature selection or techniques of explainable AI, which interrogate the trained models post hoc [246]. The feature selection algorithms reveal hidden patterns, enabling insights into disease mechanisms and identification of potential therapeutic targets. Furthermore, AI-powered predictive models utilizing mechanistically relevant features can be developed to diagnose diseases, predict treatment responses, or prognosticate patient outcomes [239]. By incorporating various patient-specific factors, clinical data, and flow cytometry profiles, these models may provide personalized and data-driven insights for precision medicine [247].

### 6.6. Omics Integration

The use of data science techniques in flow cytometry has already significantly advanced the field. However, it is essential to emphasize that the expertise of cytometrists and biologists remains vital for interpretation, validation, and translation of the findings into meaningful biological insights. Nevertheless, there is no doubt that the full integration of flow cytometry data with other omics data utilizing AI tools will provide a more comprehensive understanding of cellular function and disease processes in the future [241]. This endeavor necessitates collaboration between flow cytometrists and data science researchers to develop new analysis methods that amalgamate flow cytometry profiles with other omics perspectives, possibly using multiview learning and other cutting-edge techniques [248]. Such a partnership will generate a holistic view of biological systems, facilitating the identification of key regulatory pathways, cellular interactions, and therapeutic targets.

## 7. Cytometry Market Opportunities

Flow cytometry has become a versatile tool with widespread applications in various fields, including immunology, cancer research, stem cell research, drug discovery, and diagnostics. The increasing prevalence of chronic diseases, advancements in immune therapy and personalized medicine, and the demands for efficient diagnostic tools and effective therapies have been major drivers of growth in the flow cytometry market. Notable advancements in flow cytometry technology have improved its capabilities and expanded its applications. High-throughput flow cytometry systems, imaging flow cytometry, and integrating multiple parameters and functionalities in a single instrument have enhanced the speed, sensitivity, and multiplexing capabilities of flow cytometry, enabling more detailed analyses of cells and their subpopulations. Moreover, the integration of flow cytometry with techniques like mass spectrometry and single-cell analysis has provided deeper insights into cellular functions and interactions. The development of novel fluorochromes, antibodies, and reagents with enhanced specificity and brightness has further improved the performance and versatility of flow cytometry [249].

The global flow cytometry market was valued at $4.8 billion in 2021 and is estimated to reach $7.6 billion by 2027, with a projected CAGR of 8.0% [179]. Technological innovations across all product categories, including instruments, reagents, and software, are driving the growth of this market. Advances in reagents, instrumentation, and software have propelled the field forward. Although the high cost of instruments and the requirement for skilled personnel pose challenges, the introduction of bench-top and affordable flow cytometry platforms addresses these concerns and keeps the market attractive for investors. The market for flow cytometry will continue to grow due to the increasing demands for personalized medicine, immunotherapy, and immuno-oncology, advancements in high-dimensional analysis, and the exploration of new applications such as liquid biopsy and rare cell detection. The expanding adoption of flow cytometry in emerging markets also contributes to its market growth.

In summary, flow cytometry has emerged as a powerful tool with diverse applications, and its market continues to expand, driven by technological advancements and increasing demand across different sectors. The growth in the market for flow cytometry instruments, reagents, and software highlights the ongoing advancements and need for high-quality products and reagents to support research and diagnostic and therapeutic applications.

## 8. Next-Generation Instruments

Flow cytometry technology has evolved over many years, but even the most recent instruments contain variations of the same components and thus suffer from the same problems. For example, most instruments use lasers as simple light sources. However, the sophisticated controls that could be employed by advancing laser management have evaded flow cytometry manufacturers. This is one area that will enhance the new generation of instruments. Secondly, more advanced, more sensitive, and much faster sensors are needed if the field is to advance. Current PMTs and APDs that are used have been used for decades, and while they provide excellent results most of the time, the limitations of dynamic range, signal noise, and low sensor speed reduce the potential for detection innovation. By using very high speed sensors [250,251], next-generation instruments could produce all of the current fluorescence signals commonly collected but could, in some cases, simultaneously collect fluorescence lifetime data without compromising total fluorescence measurements. A far better understanding of signal noise could enable a much better understanding of the composite signals that are collected. In addition, if entirely digital detection methods were used, such as single photon detection, it is entirely possible that an instrument could be developed that would be genuinely quantitative in every respect. This would enable absolute signal quantification, leading to proper standardization of flow cytometry. Finally, there are currently no instruments in which either the excitation or emission system does not require alignment by the user. This is a massive component in the variability of instruments, and with the electronic and optical sophistication currently available, such a feature is desirable. The last component of next-generation flow cytometry instruments is the total integration of AI concepts, moving the field well into the mid-21st century since AI will undoubtedly to be a critical aspect of future instruments.

## 9. Conclusions

Over the past 50 years, the field of single-cell analysis has expanded from a simplistic single-parameter approach to being able to analyze well over 40 simultaneous parameters. This has been possible through the development of monoclonal antibodies, new fluorophores, the ready availability of solid-state lasers, new sensor technology, and of course, powerful desktop computers. A critical component of the success of flow cytometry has been the advancement of analytical techniques that facilitate the complex extraction of multiple populations from mixtures. As analytical techniques improved, so too did the potential for integration of these analytical processors into cell sorters, which enabled the physical separation of identified populations. With the current emergence of microfluidic systems that can facilitate sorting and separation, this will likely expand clonal selection techniques. As new sensors and better optical approaches develop, we are likely to see significant advances in flow cytometry instrumentation that will promote innovative discovery opportunities. 

## Figures and Tables

**Figure 1 cells-12-01875-f001:**
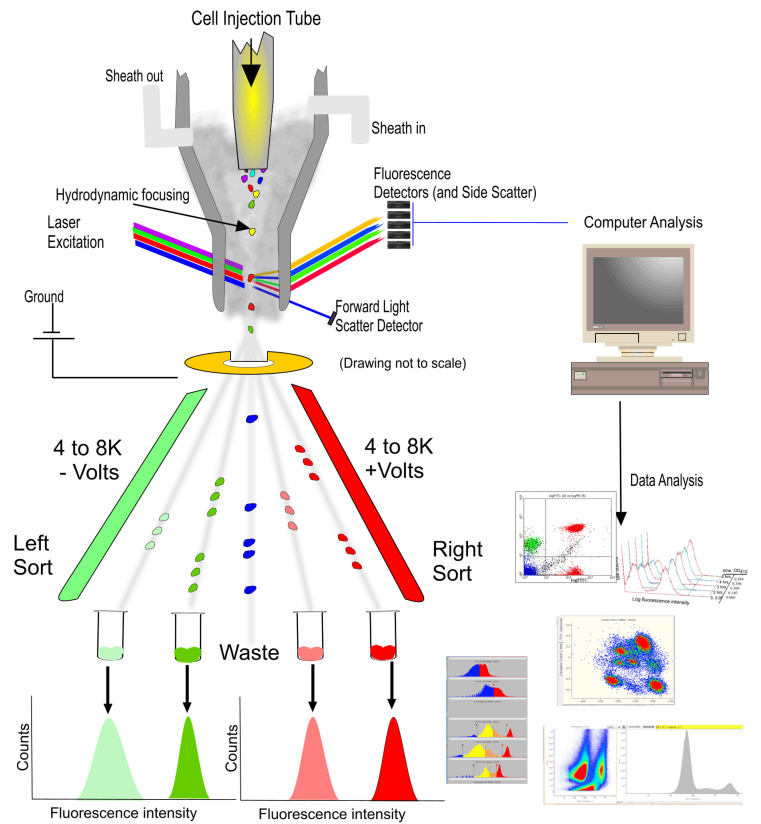
A general outline of a flow cytometer showing a sorting instrument capable of isolating individual cells. On the right side of the figure are examples of possible analyses.

**Figure 2 cells-12-01875-f002:**
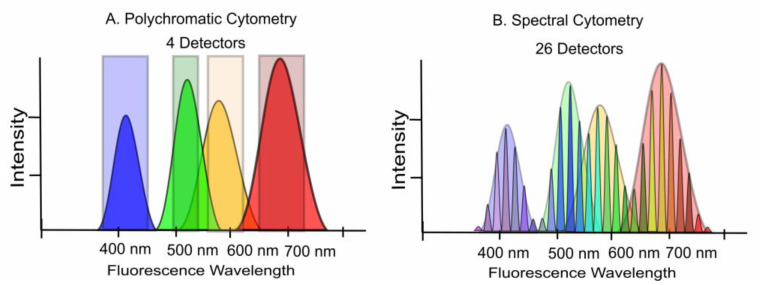
A comparison between polychromatic cytometry (**A**) and spectral cytometry (**B**). In (**A**), the fluorescence from each dye is collected by a single detector, while in (**B**), many detectors are used to collect the entire spectrum of all dyes, enabling a process called spectral unmixing that can identify each dye. There are many advantages to the spectral concept.

**Figure 3 cells-12-01875-f003:**
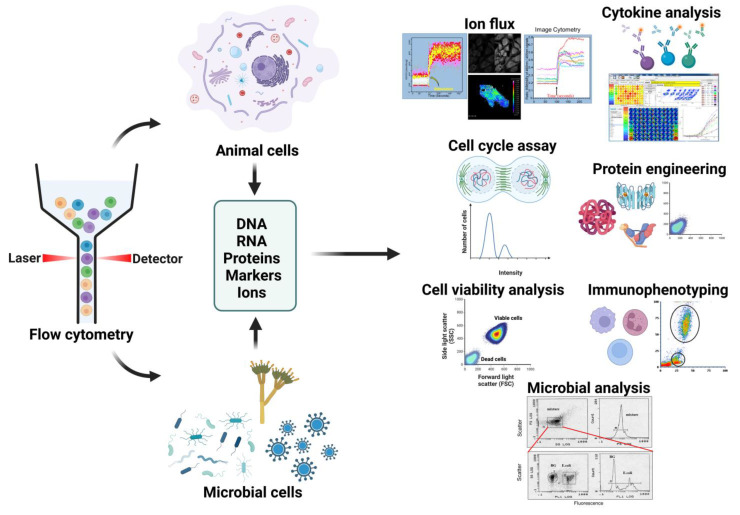
Flow cytometry applications include cellular DNA/RNA analysis, phenotyping, ion flux, microbial analysis, cytokines, plant DNA, and a variety of additional assays.

**Figure 4 cells-12-01875-f004:**
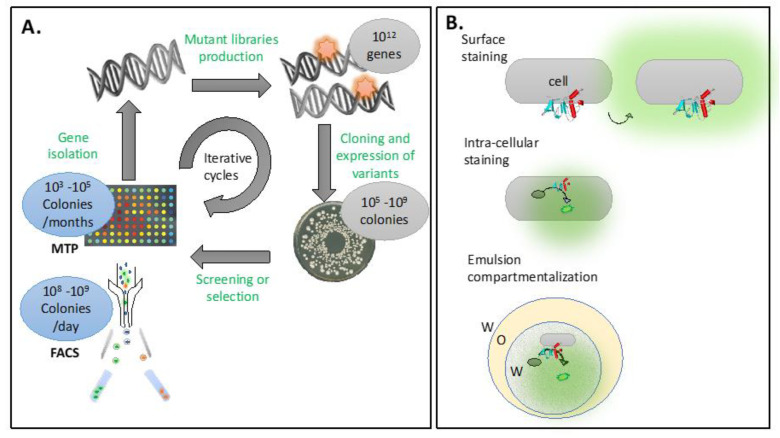
Overview of directed evolution approaches for protein engineering. (**A**) Thousands to millions of individual protein variants can be produced. (**B**) Types of cell-based assays for protein evolution compatible with flow cytometry.

**Figure 5 cells-12-01875-f005:**
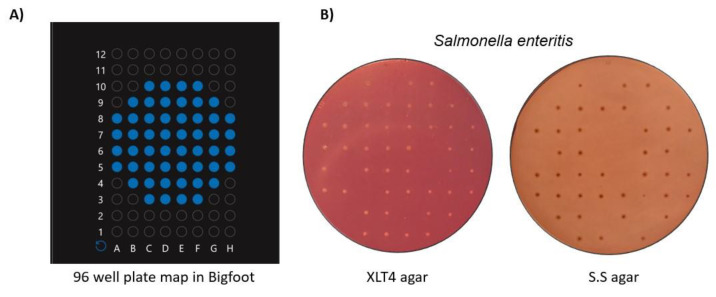
Single bacterial cell sorting using the Bigfoot cell sorter on two selective media, XLT4 and Salmonella Shigella (S.S) agar. (**A**) Screenshot of the 96-well plate map from the Bigfoot cell sorter software used as a reference to deposit single bacterial colonies at desired spots on the agar plate. (**B**) Colonies grown on XLT4 and S.S agar after 24 h of incubation.

**Table 1 cells-12-01875-t001:** Typical fluorochromes used in flow cytometry.

Fluorochrome	Excitation Spectrum	Emission Spectrum
Hoechst	350 nm	450 nm (blue)
DAPI	350 nm	450 nm (blue)
Pacific Blue	405 nm	450 nm (blue)
Brilliant Violet	405 nm	Numerous Lines
FITC	488 nm (blue)	530 nm (green)
PE	488–530 nm (green)	575 nm (orange-red)
PerCP	488 nm (blue)	675 nm (orange)
APC	633 nm (red)	660 nm (far-red)
Alexa Fluor 488	488 nm (blue)	519 nm (green)
Alexa Fluor 647	633 nm (red)	668 nm (far-red)
Alexa Fluor 700	633 nm (red)	719 nm (far-red)
Pacific Blue	405 nm (violet)	455 nm (blue)
BV421	405 nm (violet)	421 nm (blue)
BV510	405 nm (violet)	510 nm (green)
BV605	405 nm (violet)	605 nm (orange)
BV650	405 nm (violet)	650 nm (red)
BV711	405 nm (violet)	711 nm (far-red)
PE-Cy7	633 nm (red)	780 nm (near-infrared)
APC-Cy7	633 nm (red)	785 nm (near-infrared)
PerCP-Cy5.5	488 nm (blue)	695 nm (red)
Qdot 605	405 nm (violet)	605 nm (orange)
Qdot 655	405 nm (violet)	655 nm (red)
Qdot 705	405 nm (violet)	705 nm (far-red)
Qdot 800	405 nm (violet)	800 nm (near-infrared)

## Data Availability

Not applicable.

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
