# Peer review of "Flow Cytometry: The Next Revolution"

_cells, 2023, doi:10.3390/cells12141875_

Round 1

Reviewer 1 Report

The main focus of the review is development of flow cytometry techniques, especially in the field of fluorescence based cytomics. Authors also mentioned that recently developed experimental methods using droplet microfluidics would have a great potential to expand the knowledge in single cell interaction with cells or particles. Authors indicated that quick development of sensors and better optical approaches in flow cytometry technique have made great contribution in the science and will achieve further discovery in future.

The review article is well written. Important functionality and future development capacity of flow cytometry is mentioned.

There are a few minor points which should be considered.

1.       Subtitle number is confusing: There are three of no. 2. for subtitles and two of no.3. Please correct the numbers.

Line 81; 2. Fluorescence Detection Principles

Line 97; 2. The Principles of Flow Cytometry

Line 192; 2. Applications of Flow-cytometry

Line 515; 3. Polychromatic vs Spectral Cytometry

Line 539; 3. Droplet Microfluidics

Line 639¸ 4. Informatics

Line 647; 5. “Next-generation bioengineered biologics”

Line 744; 6. Market Opportunities

Line 788; 7. Next Generation Instruments

Line 714; 8. Conclusion

2. Page 8-9,  2.2 Cell viability: for phenotype analysis, cells could be fixed in order to perform intra cellular staining. There is viability staining methods that could be used before cell fixation. It should be mentioned for this part.

3. Page9,  2.3 Cell cycle analysis: Authors cited articles from 1975-1980 only. It is a period for fundamental studies in these subjects and important methods for cell cycle investigation were established then. However, there are important discoveries in this field during last decades, such as assays using BrdU, EdU. Detection of cell proliferation. Ki67 staining and CFSE labelling of cells are also commonly used. These facts could be added. Please see references on Basic Methods of Cell Cycle Analysis. Ligasová, Anna; Frydrych, Ivo; Koberna, Karel, ISSN: 1661-6596 , 1422-0067; DOI: 10.3390/ijms24043674; PMID: 36835083. International journal of molecular sciences. , 2023, Vol.24(4)

4. Page 10-11, 2,5 Protein engineering

Figure 3: Figure legends could be expanded more. At least ‘A.’ is missing from the legends.

In the following text, Figure 3 is not cited in the text. There are error marks at Line 398, 412 and 417. Please cite figures correctly.

5. Page 13, 3. Polychromatic vs Spectral Cytometry

In this text, no references are cited. Please add some references based on this text. These two review articles are closely related to the issue.

Review article, by John P Nolan The evolution of spectral flow cytometry. Cytometry A. 2022 Oct;101(10):812-817. doi: 10.1002/cyto.a.24566. Epub 2022 May 14.

Another review article by the same author of this manuscript:  J Paul Robinson Flow cytometry: past and future. Biotechniques. 2022 Apr;72(4):159-169. doi: 10.2144/btn-202-0005. Epub 2022 Apr 4.

6. Line 615 up to xxx events per second: xxx should be replaced some number?

Author Response

There are a few minor points which should be considered.

  1. Subtitle number is confusing: There are three of no. 2. for subtitles and two of no.3. Please correct the numbers.

RESPONSE: I believe we fixed this issue with new numbering

Line 81; 2. Fluorescence Detection Principles 

Line 97; 2. The Principles of Flow Cytometry

Line 192; 2. Applications of Flow-cytometry

Line 515; 3. Polychromatic vs Spectral Cytometry

Line 539; 3. Droplet Microfluidics

Line 639¸ 4. Informatics

Line 647; 5. “Next-generation bioengineered biologics”

Line 744; 6. Market Opportunities

Line 788; 7. Next Generation Instruments

Line 714; 8. Conclusion

  1. Page 8-9, 2.2 Cell viability: for phenotype analysis, cells could be fixed in order to perform intra cellular staining. There is viability staining methods that could be used before cell fixation. It should be mentioned for this part

This section was modified

  1. Page9, 2.3 Cell cycle analysis: Authors cited articles from 1975-1980 only. It is a period for fundamental studies in these subjects and important methods for cell cycle investigation were established then. However, there are important discoveries in this field during last decades, such as assays using BrdU, EdU. Detection of cell proliferation. Ki67 staining and CFSE labelling of cells are also commonly used. These facts could be added. Please see references on Basic Methods of Cell Cycle Analysis. Ligasová, Anna; Frydrych, Ivo; Koberna, Karel, ISSN: 1661-6596 , 1422-0067; DOI: 10.3390/ijms24043674; PMID: 36835083. International journal of molecular sciences. , 2023, Vol.24(4)

Agree and we enhanced this section significantly to address this with new references including the one suggested which we had not seen.

  1. Page 10-11, 2,5 Protein engineering

Figure 3: Figure legends could be expanded more. At least ‘A.’ is missing from the legends.

yes thanks - we fixed this

In the following text, Figure 3 is not cited in the text. There are error marks at Line 398, 412 and 417. Please cite figures correctly.

Fixed now - the error marks were a result of the internal linking not working

  1. Page 13, 3. Polychromatic vs Spectral Cytometry

In this text, no references are cited. Please add some references based on this text. These two review articles are closely related to the issue.

Review article, by John P Nolan The evolution of spectral flow cytometry. Cytometry A. 2022 Oct;101(10):812-817. doi: 10.1002/cyto.a.24566. Epub 2022 May 14.

Another review article by the same author of this manuscript: J Paul Robinson Flow cytometry: past and future. Biotechniques. 2022 Apr;72(4):159-169. doi: 10.2144/btn-202-0005. Epub 2022 Apr 4.

We enhanced this section significantly and added several new references

  1. Line 615 up to xxx events per second: xxx should be replaced some number?

yes fixed - it should have been 1000 -

Reviewer 2 Report

I have taken the time to review the following review manuscript: 

Flow Cytometry: The next revolution
The authors wrote an interesting manuscript. I have found several areas that could benefit from minor corrections and clarifications for an enhanced reading experience. Please find my suggestions below:

Major comments: 

1. Line 390: FACS stands for Fluorescence-Activated Cell Sorting.
2. Whole paper; It would benefit readers of a review to have a clear
focus. The abstract discusses this technology's new multi-parameter
capabilities and the conclusion discusses that. However much of the
paper focuses on several assays that have been utilized for many years
(e.g. sections 2.2, 2.3, 2.4) and other important emerging sections are
too minimal. It would strengthen the paper to expand section 4 and
discuss machine learning. New multi-parameter techniques should be
discussed instead of old and over-discussed topics (e.g. cyTOF
workflows that analyze whole blood). In general, if the paper is about
comparing flow cytometry to multiomic technologies, then the paper
should focus on the novel techniques that support that.
3. Table 1: Why aren't UV and yellow/green laser lines mentioned? The
information for the PE dye is incorrect; it's best excited with a
green laser line.
4. The informatics section requires a lot more information.

1. Line 35-36: Consider revising the phrasing as follows: “The
pioneering use of fluorescence microscopy marked a significant
departure from the traditional spectrophotometric analysis.”
2. Line 36: I recommend using a hyphen for "protein-based", joining
these words to indicate their combined meaning.
3. Line 38: This phrase could be revised for clarity: "The technique
introduced by Robert Feulgen is termed the Feulgen reaction."
4. Line 45: Please check the spelling of the word "son".
5. Line 52: The spelling of "microfluorometric" needs to be verified.
6. Line 57-59: The sentence might read better as: "His aim was to
automate the classification of cytological criteria introduced by
Papanicolaou, and he successfully integrated his instrument with
Papanicolaou's criteria."
7. Line 60: Consider adding the word "of" to the sentence: "...lookup
table for calculating the effects of the..."
8. Line 63: The sentence would sound smoother if you replace "While on
the..." with "Regarding".
9. Line 67-69: This section could be made clearer: "Moldovan's aim was
to quantify blood cells. However, due to the extremely weak signal
strength with commercially available electronics, this proved to be
challenging. Nonetheless, this publication marks the first instance of
successfully counting cells flowing in a tube."
10. Line 70: Remove the apostrophe from "1950s".
11. Line 71: Make sure to add an apostrophe in "Wallace Coulter's".
12. Line 72-74: Consider rephrasing the sentence to: "Coulter's
groundbreaking contribution to the field of cell counting came through
his revolutionary device, the Coulter Counter, which he introduced in
his sole published work."
13. Line 75: Please remove the blank space that unnecessarily
separates the section.
14. Line 84: For clarity, if you intend to mean "longer wavelength,"
it would be best to use this phrase instead of "higher wavelength" to
avoid potential confusion.
15. Line 88: Please check the spelling of "fluorochrome".
16. Line 477: Please note that FACSDiva is software, not hardware.

Please note that there are several other stylistic and language issues
that may make this review difficult for many readers. I would
recommend major English revisions if possible.

Author Response

Flow Cytometry: The next revolution
The authors wrote an interesting manuscript. I have found several areas that could benefit from minor corrections and clarifications for an enhanced reading experience. Please find my suggestions below:

Major comments: 

1. Line 390: FACS stands for Fluorescence-Activated Cell Sorting.

yes - that was a bad mistake - fixed

2. Whole paper; It would benefit readers of a review to have a clear
focus. The abstract discusses this technology's new multi-parameter
capabilities and the conclusion discusses that. However much of the
paper focuses on several assays that have been utilized for many years
(e.g. sections 2.2, 2.3, 2.4) and other important emerging sections are
too minimal. It would strengthen the paper to expand section 4 and
discuss machine learning. New multi-parameter techniques should be
discussed instead of old and over-discussed topics (e.g. cyTOF
workflows that analyze whole blood). In general, if the paper is about
comparing flow cytometry to multiomic technologies, then the paper
should focus on the novel techniques that support that.

yes the entire informatics section was written as an afterthought and based on the reviewers's very appropriate comment, we asked our colleague Dr. Bartek Rajwa in our group who is an expert in this area to re-write that entire section which is now a more significant part of the manuscript.

3. Table 1: Why aren't UV and yellow/green laser lines mentioned? The
information for the PE dye is incorrect; it's best excited with a
green laser line.

these UV dyes  have been added - and we added multiple excitation for PE as it is commonly used with the 488 laser even tho it is more efficient at higher wavelenths
4. The informatics section requires a lot more information.

As noted above - completely rewritten

1. Line 35-36: Consider revising the phrasing as follows: “The
pioneering use of fluorescence microscopy marked a significant
departure from the traditional spectrophotometric analysis.”

Accepted

2. Line 36: I recommend using a hyphen for "protein-based", joining
these words to indicate their combined meaning.

Accepted
3. Line 38: This phrase could be revised for clarity: "The technique
introduced by Robert Feulgen is termed the Feulgen reaction."

Accepted
4. Line 45: Please check the spelling of the word "son".

yes - don't know how that happened - fixed.

5. Line 52: The spelling of "microfluorometric" needs to be verified.

yes this is the spelling used in the original papers

6. Line 57-59: The sentence might read better as: "His aim was to
automate the classification of cytological criteria introduced by
Papanicolaou, and he successfully integrated his instrument with
Papanicolaou's criteria."

it does and we changed it

7. Line 60: Consider adding the word "of" to the sentence: "...lookup
table for calculating the effects of the..."

yes changed

8. Line 63: The sentence would sound smoother if you replace "While on
the..." with "Regarding".

yes it does and we did

9. Line 67-69: This section could be made clearer: "Moldovan's aim was
to quantify blood cells. However, due to the extremely weak signal
strength with commercially available electronics, this proved to be
challenging. Nonetheless, this publication marks the first instance of
successfully counting cells flowing in a tube."

much better thanks

10. Line 70: Remove the apostrophe from "1950s".

yes removed
11. Line 71: Make sure to add an apostrophe in "Wallace Coulter's".

yes done
12. Line 72-74: Consider rephrasing the sentence to: "Coulter's
groundbreaking contribution to the field of cell counting came through
his revolutionary device, the Coulter Counter, which he introduced in
his sole published work."

yes done

13. Line 75: Please remove the blank space that unnecessarily
separates the section.

we had some trouble removing this space - it seems WORD appears to be the controller here - so maybe this has to be removed in copyediting

14. Line 84: For clarity, if you intend to mean "longer wavelength,"
it would be best to use this phrase instead of "higher wavelength" to
avoid potential confusion.

yes better

15. Line 88: Please check the spelling of "fluorochrome".

correct

16. Line 477: Please note that FACSDiva is software, not hardware.

yes thanks for that - fixed   Comments on the Quality of English Language

Please note that there are several other stylistic and language issues
that may make this review difficult for many readers. I would
recommend major English revisions if possible.

because there were several authors, the language can be a little different - we edited some of those sections to try to make it more consistent